# Randomized Crossover Trial Evaluating Detoxification of Tobacco Carcinogens by Broccoli Seed and Sprout Extract in Current Smokers

**DOI:** 10.3390/cancers14092129

**Published:** 2022-04-24

**Authors:** Julie E. Bauman, Chiu-Hsieh Hsu, Sara Centuori, Jose Guillen-Rodriguez, Linda L. Garland, Emily Ho, Megha Padi, Vignesh Bageerathan, Lisa Bengtson, Malgorzata Wojtowicz, Eva Szabo, H.-H. Sherry Chow

**Affiliations:** 1Department of Medicine, Division of Hematology/Oncology, University of Arizona (UA) and UA Cancer Center, Tucson, AZ 85724, USA; smb4@arizona.edu (S.C.); lgarland@uacc.arizona.edu (L.L.G.); schow@arizona.edu (H.-H.S.C.); 2Department of Medicine, Division of Hematology/Oncology, George Washington (GW) University and GW Cancer Center, Washington, DC 20037, USA; 3Department of Epidemiology and Biostatistics, UA and UA Cancer Center, Tucson, AZ 85724, USA; pchhsu@arizona.edu; 4Biostatistics and Bioinformatics Shared Resource, UA Cancer Center, Tucson, AZ 85724, USA; jguillen@arizona.edu (J.G.-R.); mpadi@arizona.edu (M.P.); vbageerathan@arizona.edu (V.B.); 5Linus Pauling Institute, College of Public Health and Human Sciences, Oregon State University, Corvallis, OR 97331, USA; emily.ho@oregonstate.edu; 6Department of Molecular and Cellular Biology, UA, Tucson, AZ 85724, USA; 7Division of Cancer Prevention, National Cancer Institute, Bethesda, MD 20892, USA; lisa.bengtson@nih.gov (L.B.); malgorzata.wojtowicz@nih.gov (M.W.); szaboe@mail.nih.gov (E.S.)

**Keywords:** sulforaphane, glucoraphanin, detoxification, tobacco carcinogens, clinical trial, broccoli seed and sprout extract, smokers

## Abstract

**Simple Summary:**

Cruciferous vegetable consumption is associated with reduced risk of tobacco-related cancers. Broccoli seed and sprout extract (BSSE) is rich in the isothiocyanate glucoraphanin, which is hydrolyzed to the phytochemical sulforaphane, a potent inducer of cytoprotective enzymes. This randomized, crossover trial evaluated detoxification of tobacco carcinogens by two doses of the BSSE Avmacol^®^ in 49 otherwise healthy tobacco smokers. The higher dose (296 µmol of glucoraphanin daily) significantly upregulated detoxification of benzene, acrolein, and crotonaldehyde.

**Abstract:**

Consumption of cruciferous vegetables, rich in the isothiocyanate glucoraphanin, is associated with reduced risk of tobacco-related cancers. Sulforaphane, released by hydrolysis of glucoraphanin, potently induces cytoprotective phase II enzymes. Sulforaphane decreased the incidence of oral cancer in the 4NQO carcinogenesis model. In residents of Qidong, China, broccoli seed and sprout extracts (BSSE) increased detoxification of air pollutants benzene and acrolein, also found in tobacco smoke. This randomized, crossover trial evaluated detoxification of tobacco carcinogens by the BSSE Avmacol^®^ in otherwise healthy smokers. Participants were treated for 2 weeks with both low and higher-dose BSSE (148 µmol vs. 296 µmol of glucoraphanin daily), separated by a 2-week washout, with randomization to low-high vs. high-low sequence. The primary endpoint was detoxification of benzene, measured by urinary excretion of its mercapturic acid, SPMA. Secondary endpoints included bioavailability, detoxification of acrolein and crotonaldehyde, modulation by *GST* genotype, and toxicity. Forty-nine participants enrolled, including 26 (53%) females with median use of 20 cigarettes/day. Low and higher-dose BSSE showed a mean bioavailability of 11% and 10%, respectively. Higher-dose BSSE significantly upregulated urinary excretion of the mercapturic acids of benzene (*p* = 0.04), acrolein (*p* < 0.01), and crotonaldehyde (*p* = 0.02), independent of *GST* genotype. Retention and compliance were high resulting in early study completion. In conclusion, BSSE significantly upregulated detoxification of the tobacco carcinogens benzene, acrolein, and crotonaldehyde in current tobacco smokers.

## 1. Introduction

Habitual tobacco use is the major risk factor for human papillomavirus (HPV)-negative head and neck squamous cell carcinoma (HNSCC) [1]. Long-term survival following curative-intent treatment of tobacco-related HNSCC is compromised by an alarming, 3–6% annual rate of second primary tumor (SPT) development within the upper aerodigestive tract, including lung, esophageal, or head and neck cancer [2,3,4,5]. Indeed, SPTs represent the leading cause of mortality after the treatment of early-stage HNSCC [6,7]. Although tobacco cessation reduces the incidence of SPTs, hazard reduction is not observed for 5 years and risk never returns to baseline [4,8]. Even though the diagnosis of HNSCC represents a teachable moment, and the majority of patients express interest in tobacco cessation, approximately 50–70% of current smokers continue to smoke after diagnosis [9,10,11]. Moreover, among the 30–50% who make a quit attempt, the majority relapse even when provided with evidence-based tobacco cessation support [12,13,14]. Additional primary and secondary prevention strategies are needed.

The possibility of chemoprevention for HNSCC was raised in 1953 when Slaughter and colleagues first described epithelial field cancerization, or multicentric histologic abnormalities within the grossly normal mucosa surrounding an index tumor [15]. This so-called “condemned mucosa,” due to chronic exposure to tobacco and other environmental carcinogens, gives rise to SPTs throughout the upper aerodigestive tract. When epidemiologic studies found that decreased risk for both HNSCC and SPTs is associated with diets rich in fruits and vegetables [16,17,18,19,20], the first chemoprevention strategies focused on pharmacologic doses of derivative micronutrients such as vitamin A. Preclinical models supported retinoids, synthetic vitamin A analogues, as modulators of oral environmental carcinogenesis [21,22]. Indeed, high-dose isotretinoin reversed oral premalignant lesions (OPLs) and prevented SPTs in two landmark HNSCC chemoprevention studies [23,24]. However, toxicity made regular use impracticable and SPT incidence returned to baseline within 3 years of treatment discontinuation. In a follow-up trial, low-dose isotretinoin was well-tolerated but did not prevent SPTs [25]. An effective, tolerable chemopreventive agent against tobacco-related HNSCC and SPTs represents an unmet global need.

Reduced risk for both HNSCC and SPTs is associated with consumption of the *Brassica* family of cruciferous vegetables, such as broccoli, kale, and cabbage [16,17,18,19,20]. Isothiocyanates, stress-response phytochemicals derived from the glucosinolates in Cruciferae, potently induce cytoprotective phase II enzymes [26], which enhance detoxification of benzene, aldehydes, and polycyclic aromatic hydrocarbons found in tobacco smoke [27,28]. The phytochemical sulforaphane (SFN), produced when its precursor glucoraphanin (GR) is hydrolyzed by the plant enzyme myrosinase released during food preparation or chewing, is responsible for 80% of phase II enzyme induction by broccoli extract [29]. SFN disrupts the inhibition of the nuclear factor (erythroid-derived 2)-like 2 (NRF2) transcription factor by Kelch-like ECH-associated protein 1 (KEAP1), freeing NRF2 to translocate to the nucleus and bind antioxidant response elements (AREs) in the promoter regions of target genes, including enzymes such as NAD(P)H quinone oxidoreductase 1 (NQO1) that detoxify chemical carcinogens [30,31]. Immature broccoli sources, such as broccoli seeds and sprouts, are highly concentrated in GR, making broccoli seed and sprout extracts (BSSE) intriguing candidates for chemoprevention of tobacco-related cancers [29,32]. We have shown that SFN decreases the incidence and size of 4NQO-induced HNSCC in mice and that SFN-rich BSSE upregulates *NQO1* gene transcripts in the oral mucosa of healthy volunteers [33]. Various BSSE formulations have been well-tolerated in human studies [31,33,34,35,36]. Notably, a BSSE containing both GR and SFN sustainably increased detoxification of the air pollutants benzene and acrolein, also found in tobacco smoke [37], motivating the clinical trial reported here: a randomized, crossover trial evaluating the detoxification of tobacco carcinogens by two doses of BSSE in otherwise healthy, current smokers.

## 2. Participants and Methods

### 2.1. Participants

This single-center study was conducted at the University of Arizona Cancer Prevention Clinic (Tucson, AZ, USA) and sponsored by the U.S. National Cancer Institute (NCI) Division of Cancer Prevention through the Cancer Prevention Agent Development Program: Early Phase Clinical Research (HHSN2612012000311). The protocol was approved by the NCI Central Institutional Review Board and carried out in accordance with the Declaration of Helsinki and Good Clinical Practice. All participants provided written informed consent. The study was registered with clinicaltrials.gov (NCT03402230). Key inclusion criteria included: male or female aged ≥18 years; current tobacco smoker with ≥20 pack years of self-reported smoking exposure and a current average use of ≥10 cigarettes/day; Karnofsky performance status ≥70%; adequate organ and marrow function. Key exclusion criteria included: history of invasive cancer within the past 2 years; regular use of supraphysiologic steroid doses; uncontrolled intercurrent illness.

The trial was conducted according to an open label, randomized, crossover design evaluating the detoxification of tobacco carcinogens by two doses of BSSE in otherwise healthy, current heavy smokers. Participants were randomized to a low-high or high-low dose sequence and served as their own controls. The primary objective was to determine if BSSE increased detoxification of benzene, as measured by change in urinary excretion of the mercapturic acid of benzene, S-phenyl mercapturic acid (SPMA). Secondary objectives included: to determine if BSSE increased the detoxification of acrolein and crotonaldehyde, as measured by change in urinary excretion of their respective mercapturic acids, 3-hydroxypropyl mercapturic acid (3-HPMA) and 3-hydroxy-1-methylpropyl mercapturic acid (3-HMPMA); to measure the bioavailability of the BSSE Avmacol^®^ as measured by urinary SFN metabolites; to determine if BSSE upregulates buccal expression of NRF2 target genes including NQO1; to evaluate for a dose-response relationship between effective SFN dose, the detoxification of tobacco carcinogens, and the buccal expression of NRF2 target genes; and to describe the toxicity of low and higher-dose BSSE according to the NCI Common Terminology Criteria for Adverse Events (CTCAE) version 4.0. The exploratory objective was to determine if the glutathione S-transferases (GST) alleles, *GSTM1* and *GSTT1*, are important genetic modulators of detoxification of tobacco carcinogens. We planned to accrue 61 eligible participants to initiate agent intervention. With an anticipated attrition rate of ≤33%, as observed in chemoprevention studies in the same population ([38] and NCT02348203), we expected at least 41 participants with evaluable endpoint data. With a minimum evaluable sample size of 41 and an overall significance level of 5%, there would be at least 80% power to detect an effect size of 0.50 standard deviations (SD), in line with an effect size of 0.45 SD observed in a similar study [39].

### 2.2. Study Agent and Treatment Plan

BSSE was delivered as the commercially available nutraceutical, Avmacol^®^, manufactured under good manufacturing practice (GMP) standards by Nutramax Laboratories, Inc. The original Avmacol^®^ formulation is comprised of GR-rich broccoli seed extract, freeze-dried broccoli sprouts for the myrosinase source, ascorbic acid, and inert excipients for tablet formation (carboxymethylcellulose, silicon dioxide, and microcrystalline cellulose). Manufacturing specifications require each tablet to contain at least 13 mg GR with sufficient myrosinase activity to generate at least 5 mg SFN. The same lot of Avmacol^®^ (RD0416-03, manufactured 04/2016) was used throughout the study; each tablet contained 16 mg (37 µmol) GR with SFN conversion of 6 mg (34 µmol). Stability testing was performed every 6 months throughout the study and confirmed potency, indicating a shelf life of more than 60 months at ambient temperature.

In a 2 × 2 crossover design, each participant was treated with both low and higher-dose BSSE, comprised of 4 vs. 8 Avmacol^®^ tablets (148 µmol vs. 296 µmol of GR) daily. Each intervention period was 2 weeks, separated by a 2-week washout to mitigate carry-over effect. Randomization was to a low-high vs. high-low dose sequence as depicted in Figure 1. With an estimated conversion rate of 35%, due to the activity of both the co-delivered myrosinase as well as the β-thioglucosidases occurring in the human gut microbiome, the low vs. higher doses were estimated to represent approximately 50 vs. 100 µmol of bioavailable SFN [40]. BSSE was taken each evening. Overnight urine, from approximately 6:00 p.m. to 6:00 a.m., was collected 4 times during the study: at the baseline of each treatment period before the first dose of BSSE; and after the final evening BSSE dose of each treatment period. Participants self-collected urine into opaque jugs containing ascorbic acid and brought the jug to that morning’s appointment, where volume was measured and two 10-mL aliquots frozen at −80 °C. Buccal cells were collected by cytobrush and placed into RNALater in 1-mL cryovials by a trained coordinator at the baseline and end of each treatment period, then immediately frozen and stored at −80 °C as previously described [41].

### 2.3. Biomarker Analysis

#### 2.3.1. Carcinogen Metabolites

Urinary mercapturic acids of benzene, acrolein, and crotonaldehyde were quantified by sensitive and specific liquid chromatography-tandem mass spectrometry (LC-MS/MS) assays previously used in smokers’ cohorts, with minor modifications [42,43]. Briefly, for SPMA, an aliquot of urine was mixed with the internal standard (SPMA-d5) and extracted by solid phase extraction prior to LC-MS/MS analysis. For 3-HPMA and 3-HMPMA, an aliquot of urine was mixed with the internal standards (3-HPMA-d3, 3-HMPMA-13C3-15N, respectively) and an aliquot of ammonium acetate prior to LC-MS/MS analysis. LC-MS/MS analysis was performed on a Thermo TSQ Quantum Ultra triple quad mass spectrometer with Surveyor HPLC system. The analytes were chromatographically separated using a mobile phase consisting of ammonium acetate and methanol and a Synergi MAX RP column and detected in the negative ESI mode with multiple reaction monitoring utilizing the following mass transitions: 3-HPMA (220 > 91), 3-HMPMA (234 > 105), 3-HPMA-d3 (223 > 91), 3-HMPMA-13C3-15N (238 > 105), SPMA (238 > 109), and SPMA-d5 (243 > 114). The calibration curves for 3-HPMA and HMPMA were linear over the urine concentration range of 0.07–150 nmol/mL. The calibration curve for SPMA was linear over the urine concentration range of 0.2–420 pmol/mL. Urinary carcinogen metabolite levels were normalized by urinary creatinine concentrations determined by a creatinine assay kit (Diazyme laboratories). All baseline and post-intervention samples collected from an individual participant were analyzed together in the same batch.

#### 2.3.2. SFN Metabolites

The mean conversion of GR to SFN was calculated as the percent of the GR dose excreted as total SFN metabolites in overnight urine, with normalization on a molar equivalence basis. Matched baseline and end-of-treatment urine samples were analyzed for SFN, SFN-Cys (299 > 114), SFN-GSH (485 > 179), SFN-CG (356 > 114), and SFN-NAC (341.1 > 114) in duplicate following a 10-μL injection. Instrumentation and LC-MS/MS conditions were the same as used previously [44,45]. SFN metabolites were normalized to urine creatinine to account for variation in collection time and volume, as previously described [37].

#### 2.3.3. Buccal Cell mRNA

Total mRNA was isolated from the stored cryovials using Quick-RNA MiniPrep Plus (Zymo Research Corporation, Irvine, CA, USA). mRNA concentrations were quantified by the NanoDrop spectrophotometer (Thermo Fisher Scientific, Waltham, MA, USA) and verified by Qubit fluorometric quantification, as previously described [41]. Buccal mRNA was analyzed by the NanoString nCounter^®^ PanCancer panel customized to add additional NRF2-dependent and independent target genes including: *AKR1B10, AKR1C1, CAT, CEACAM1, DEFB, FCER2, GCLC, GCLM, HMOX1, HSPA4, HSPB1, LYVE1, MIF, NFE2L2, NQO1, PPIA, SLC7A11, SMPD1, SOD2,* and *TRAF6.* Between 100–300 ng of the purified mRNA was used in the NanoString analysis depending on quality and availability. Briefly, samples were hybridized overnight with the Gene Expression Code Set for the human PanCancer IO 360 panel and custom Panel-Plus gene spike-in (NanoString Technologies, Seattle, WA, USA) at 65 °C for 19 h. Further purification and binding of the hybridized probes to the optical cartridge were performed on the automated nCounter Prep Station using the high sensitivity setting, and the cartridge was scanned on the nCounter Digital Analyzer at maximum resolution (555 FOV). Raw counts from each gene per sample were imported into the nSolver Analysis Software and overall assay performance was assessed through evaluation of built-in positive controls, binding density evaluation, and number of probes above background. Batch calibration was used to harmonize data generated in different runs, with panel standards as references. Background thresholding was applied using the mean + 2 SD. Normalization was carried out with respect to the geometric mean of the positive controls, and with respect to housekeeping genes using the Normalization Module of nCounter Advanced Analysis. Log_2_-transformed normalized expression values were used for all further analysis.

#### 2.3.4. GSTM1 and GSTT1 Genotypes

Genomic DNA was isolated from circulating blood lymphocytes collected at the baseline clinic visit. *GSTM1* and *GSTT1* were genotyped using standard methods as previously described [46]. Primers were ordered from Eurofins with fluorophores for visualization using a capillary electrophoresis system and genetic material was amplified using PCR [46,47]. Samples were diluted 1/50 before being run on 3730 DNA sequencer (Applied Biosystems). Resulting .fsa files were visualized using GeneMarker (SoftGenetics). Samples in which amelogenin did not amplify were not scored but re-amplified again. Presence of a peak over 10K RFUs at 274 bp is *GSTM1* positive and presence of a peak over 10K RFUs at 456 bp is *GSTT1* positive. Absence of the peaks with positive amelogenin amplification is scored as null.

### 2.4. Statistical Analysis

For changes in the urinary excretion of SFN as well as the mercapturic acids of tobacco carcinogens, concentrations were normalized to creatinine then log-transformed due to high right skew. The changes following low- and higher-dose BSSE were determined independently. The ratio of the geometric means between post and pre concentrations and the associated 95% CI were derived from log-normal regression. Spearman correlation coefficient was calculated to evaluate the correlation between effective SFN dose, as measured by urinary SFN metabolites, and carcinogen detoxification. Changes in buccal expression of gene transcripts associated with effective SFN dose were computed using the Pearson correlation between log_2_(fold change in expression) and log_2_(fold change in creatinine-normalized SFN metabolite concentration) for each participant. Functional enrichment was evaluated by using ClusterProfiler to perform overrepresentation analysis on the top thirty genes most positively or negatively correlated with effective SFN dose. The background universe was set to only the 770 genes probed in the customized NanoString panel. Linear mixed effects models were fit to determine whether there was a dose-response relationship between effective SFN dose, carcinogen detoxification, and the change in NRF2 target gene transcripts. To explore whether the *GSTM1* and *GSTT1* genotypes were associated with detoxification, two-sided, two-sample t tests were performed. Bonferroni correction was used to correct for multiple comparisons. McNemar’s test was performed to compare the frequency of each specific AE between low and higher-dose treatment. The SAS 9.4 software package was used for all statistical analyses.

## 3. Results

### 3.1. Study Population

A total of 49 participants were enrolled from January 2018 to November 2019 as depicted in the Consolidated Standards of Reporting Trials (CONSORT) diagram (Figure 2). One hundred-and-twelve potential participants were telephone screened; 58 did not proceed to a screening visit with the most common reasons including insufficient tobacco history (*n* = 21), uninterested/declined (*n* = 20), cancellation or no-show to screening visit (*n* = 7), and not currently smoking (*n* = 5). Five of 54 were disqualified at the screening visit due to inability to attend all scheduled study visits (*n* = 2), uncontrolled hypertension (*n* = 1), non-consent for biological samples (*n* = 1), and clinically significant alanine transferase (ALT) value (*n* = 1). Due to high retention and compliance, the trial was completed early; 48 of 49 participants completed all study treatment and biospecimen collections and were evaluable for the primary endpoint.

Baseline characteristics of all 49 enrolled participants are presented in Table 1. Twenty-six of 49 participants (53%) were female, median age was 57 years, and median tobacco exposure was 36 pack-years with a median daily use of 20 cigarettes. The racial and ethnic composition of the trial cohort paralleled the tobacco-smoking populations of Arizona, where Hispanics have lower smoking rates than non-Hispanic whites [48]. 

### 3.2. Study Endpoints

#### 3.2.1. Bioavailability 

Participants demonstrated a significant increase in SFN metabolites on both low-dose and higher-dose treatment with BSSE. The median baseline urinary excretion of SFN metabolites was 0.009 µmol/mg Cr (interquartile range (IQR): 0.01, 0.03) during period 1 and 0.03 µmol/mg Cr (IQR: 0.02, 0.04) during period 2 (*p* = 0.02, indicating a statistically significant but negligible carryover effect). During low dose exposure, the geometric mean of SFN metabolites increased from 0.03 (0.02, 0.05) to 13.3 (7.9, 22.6) µmol/mg Cr, representing a post:pre ratio of 267.9 (264.7, 826.0; *p* < 0.0001). During higher-dose exposure, the geometric mean of SFN metabolites increased from 0.03 (0.02, 0.04) to 26.09 (16.2, 42.0) µmol/mg Cr, representing a post:pre ratio of 990.0 (592.2, 1655.1; *p* < 0.0001). The median duration of overnight urine collection was 11 h (range 5.8–13.7 h). The mean bioavailability of BSSE during the first 11 h, as measured by total SFN metabolites in overnight urine divided by the total administered GR dose, normalized on a molar equivalence basis, was 11% (range 0.04–59%; SD 11%) on low-dose and 10% (range 0.1–46%; SD 10%) on higher-dose treatment. A dose-response relationship between dose (low vs. high) and urinary excretion of SFN metabolites was observed (*p* < 0.01), derived from a linear mixed effects model with random intercepts accounting for within-subject correlation after adjusting for the period effect.

#### 3.2.2. Carcinogen Detoxification

The summary of changes in urinary excretion of the detoxified products of the three specified tobacco carcinogens is presented in Table 2. Higher-dose BSSE significantly upregulated detoxification of benzene, acrolein, and crotonaldehyde in current heavy smokers. Low dose BSSE significantly upregulated detoxification of benzene, however not acrolein or crotonaldehyde.

The median baseline urinary excretion of the mercapturic acid of benzene, SPMA, was 8.1 pmol/mg Cr (IQR 4.8, 13.6) in period 1 and 8.0 pmol/mg Cr (IQR 4.4, 11.6) during period 2 (*p* = 0.35, indicating no carryover effect). During higher-dose BSSE, the geometric mean of SPMA increased from 7.6 (6.2, 9.4) to 9.1 (7.3, 11.3) pmol/mg Cr, representing a post:pre ratio of 1.2 (1.0, 1.4; *p* = 0.04). The geometric mean of 3-HPMA increased from 9934.5 (8008.0, 12,324.4) to 11,032.1 (9161.1, 13,285.2) pmol/mg Cr, representing a post:pre ratio of 1.3 (1.1, 1.5; *p* < 0.01). The geometric mean of 3-HMPMA increased from 10,808.6 (8947.3, 13057.0) to 12,795.1 (10,922.9, 14,988.1) pmol/mg Cr, representing a post:pre ratio of 1.2 (1.0, 1.4; *p* = 0.02). During higher-dose BSSE treatment, there was modest positive correlation between effective SFN dose and detoxification of benzene (rho = 0.21), acrolein (rho = 0.15), and acetaldehyde (rho = 0.16).

#### 3.2.3. Buccal Cell Gene Expression

Gene expression in response to treatment exhibited substantial variability. To explore this further, genes were ranked by their Pearson correlation with effective SFN dose (Appendix A). The NRF2-dependent genes *NQO1* and *GCLM* exhibited correlation coefficients of 0.38 and 0.34, respectively, and were ranked first and second out of all 770 genes included in the panel. The top 30 genes positively correlated with effective SFN dose were evaluated for functional enrichment using Gene Ontology (GO) biological processes (BP). Although the enrichment p-values were not significant after adjusting for multiple testing, the highest-ranked biological processes were “cellular amino acid metabolic process” (GO:0006520), “organonitrogen compound metabolic process” (GO:1901564), “cellular response to stress” (GO:0033554), and “leukocyte apoptotic process” (GO:0071887) (Appendix A).

#### 3.2.4. GSTT1 and GSTM1 Genotypes

Two *GST* alleles mechanistically associated with the NRF2 detoxification pathways, *GSTM1* and *GSTT1*, were evaluated for association with the primary endpoint on higher-dose BSSE. Twenty-five of 48 (52%) participants were *GSTM1* positive. Forty of 48 (84%) participants were *GSTT1* positive. Participants who were null vs. positive for *GSTT1* had significantly less baseline excretion of SPMA (*p* = 0.02). Neither allele was associate with baseline excretion of 3-HPMA or 3-HMPMA. The upregulation of benzene, acrolein, or crotonaldehyde detoxification was independent of genotype in this sample.

#### 3.2.5. Linear Regression Analysis for SPMA Detoxification

An exploratory linear regression analysis was performed for changes in log-transformed SPMA only during the higher-dose period, in order to determine the variables most strongly associated with carcinogen detoxification. First, variables including age, sex, race, ethnicity, body mass index, *GST* genotypes, and buccal gene expression changes were evaluated for univariate association with changes in SPMA. Variables with an unadjusted *p*-value of <0.01 were then included in the adjusted analysis (Appendix A). The variables most strongly associated with benzene detoxification were changes in the buccal mRNA expression of laminin-5γ-2 (LAMC2; *p* < 0.001), interferon-induced transmembrane protein 1 (IFITM1; *p* = 0.018), and glypican 4 (GPC4; *p* = 0.027).

#### 3.2.6. Safety and Toxicity 

Adverse events (AEs) attributed to protocol treatment are summarized in Table 3 by low-dose or higher-dose exposure. Twenty-nine of 49 (59%) participants reported at least one mild, grade 1 or 2 AE; all 29 (100%) noted a gastrointestinal AE. No grade 3 or higher AE were reported. The most common AE were loose stool not meeting CTCAE criteria for diarrhea (24%) or diarrhea (22%), all Grade 1. Ten of 49 (20%) participants experienced mild abdominal pain or cramping typically associated with loose stool or diarrhea during higher-dose exposure; 4 AE were grade 1 and 6 were grade 2. The incidence of mild abdominal pain and diarrhea was significantly greater during higher-dose treatment (*p* < 0.01 and 0.02, respectively); these expected AE have been observed in previous studies and are likely attributable to the high fiber content of BSSE supplements [31,33,34,35,36]. However, no participant withdrew from the study due to a treatment-related AE. One participant randomized to the high-low dose sequence withdrew from the study during the washout period after experiencing an unrelated myocardial infarction.

## 4. Discussion

Cancers of the upper aerodigestive tract, including HPV-negative HNSCC, are associated with chronic exposure to environmental carcinogens, chiefly those present in tobacco smoke including benzene, aldehydes, and polycyclic aromatic hydrocarbons. Resultant epithelial field cancerization manifests as a high rate of morbid and often-fatal SPTs following curative treatment of an index HNSCC. Even when successful, tobacco cessation only partially mitigates SPT risk. Despite a 50-year history of rational chemoprevention efforts against HNSCC, no agent has been found to be sufficiently safe and effective to become standard of care. The present clinical trial was motivated by three key findings: First, SFN prevents HNSCC in the 4NQO model of oral carcinogenesis, a preclinical model with high molecular fidelity to human HPV-negative HNSCC [49]. Second, SFN is bioactive in the oral epithelium of healthy volunteers, upregulating NRF2 target genes [33]. Finally, daily ingestion of a different BSSE formulation containing 600 μmol GR and 40 μmol SFN resulted in the sustained detoxification of the outdoor air pollutants benzene and acrolein, also found in tobacco smoke, in residents of Qidong, China over a 12-week intervention period [37].

The key finding from this randomized, crossover clinical trial is that the BSSE Avmacol^®^, composed of GR and active myrosinase, upregulated the detoxification of the tobacco carcinogens benzene, acrolein, and crotonaldehyde, in otherwise healthy smokers when administered at the higher dose over a 2-week exposure period. Although the mean bioavailability in this trial population was approximately half that described in a formal bioavailability study following a single dose of 8 tablets [40], SFN exposure was sufficient for bioactivity. The higher 8-tablet BSSE dose, containing 296 µmol GR with a mean bioavailability of 30 µmol SFN over the first 11 h, was more effective; however, a formal dose-response relationship between effective SFN dose and detoxification response was not observed. The explanation is likely two-fold. First, the BSSE product used in the Qidong air pollution study contained both GR and SFN, the latter 70–80% bioavailable as SFN metabolites and buttressing the lower quartile of dose [37]. Avmacol^®^ co-administers GR and myrosinase, however still requires in situ conversion of GR to SFN resulting in higher variability in effective SFN dose. Second, unlike the Qidong study where all participants were exposed to the same concentrations of outdoor air pollutants due to breathing the same regional air, the current trial was conducted in the setting of variable carcinogen exposures dependent upon the brand and number of cigarettes/day smoked by individual participants. Notably, a dose-response relationship was observed in a follow-up Qidong air pollution study evaluating three doses of BSSE (the original dose of 600 μmol GR and 40 μmol SFN; one-half dose; one-fifth dose) in upregulating detoxification of air pollutants, where only the original dose corresponding to a median 24-h urinary output of 24.6 μmol SFN metabolites was significantly different from placebo control [50]. In the present study in tobacco smokers, the higher inter-individual variability in both SFN metabolites and carcinogen exposures may have obscured the expected dose-response relationship.

A frequent and important question in green chemoprevention, the use of plants or their simple extracts as cancer prevention agents, is whether a bioactive dose of protective phytochemicals could be achieved by normal diet. The GR content of market stage broccoli is highly variable, and is largely dependent upon genotype although environmental conditions such as field location, year, drought conditions, and disease pressure also play a role [51,52]. In a study of 31 fresh, uncooked (*Brassica oleracea* var. *italica*), market stage broccoli heads purchased at supermarkets in the Baltimore region, a mean GR content of 0.4 µmol (range 0.005 to 1.1 µmol) per gram was observed [53]. On average, roughly 28 ounces of market stage broccoli would contain a similar GR content as higher-dose Avmacol^®^ used in the present study. As broccoli sprouts contain 10–100 times the level of GR as mature plants, and are a rich source of myrosinase activity, their consumption may be more practical [32].

The correlations between SFN metabolites and gene expression changes in buccal cells, a surrogate epithelial tissue of interest for studying the biologic effects of purported chemopreventive agents in tobacco smokers [41], also found high inter-individual variability. Nonetheless, this exploratory analysis found over-representation of genes involved in cellular amino acid and organonitrogen compound metabolism, cellular response to stressful stimuli, and intrinsic apoptosis to correlate with effective SFN dose. Upregulation of the cytoprotective antioxidant response, including detoxification enzymes such as *NQO1* and *GCLM*, is an expected, NRF2-dependent mechanism of SFN, suggesting the relevance of this analytic approach. In preclinical models, SFN also decreases chronic inflammation through inhibition of NF-κB and enhances intrinsic apoptosis via epigenetic mechanisms [54,55]. Of interest, multivariate analysis also identified changes in buccal expression of *IFITM1*, an interferon-inducible protein with antiproliferative and anti-inflammatory functions, to be significantly associated with benzene detoxification [56,57]. These exploratory findings are in line with the hypothesis that SFN may counter the dysregulation of detoxification, inflammation, and apoptosis present during carcinogenesis via NRF2-dependent and -independent mechanisms.

As observed in prior studies, including the Qidong air pollution study, the *GSTT1* but not the *GSTM1* genotype influenced baseline benzene metabolism, as measured by SPMA excretion [37,39]. Specifically, participants null for the *GSTT1* allele showed significantly less baseline excretion of SPMA compared to those who were positive. However, the upregulation of benzene detoxification in response to study treatment occurred independent of *GST* polymorphisms. Of interest, a similar trial evaluating the effect of the isothiocyanate 2-phenethyl isothiocyanate (PEITC) on benzene and acrolein detoxification in smokers found those with single or double deletion polymorphisms in *GSTT1* and *GSTM1* disproportionately benefitted from treatment [39], an effect that may have been visible due to the larger sample size of 82 participants. As the deletion polymorphisms of *GSTT1* and *GSTM1* are associated with increased risk for HNSCC and interact multiplicatively with tobacco exposure [58], chemoprevention with isothiocyanates may be most beneficial in double null individuals.

As observed in previous trials of various BSSE, toxicity was predominantly gastrointestinal, manifesting as grade 1 loose stool or diarrhea associated with mild, crampy abdominal pain. These AEs were more common during higher-dose treatment, were expected due to the fiber content, and did not impede compliance. In fact, contrary to our previous studies in the same population ([38] and NCT02348203), we observed only a 3% vs. 33% dropout rate. Accordingly, high retention and compliance resulted in early completion of the trial at 49 rather than 61 enrolled participants.

While representative of the tobacco-smoking population of Arizona, the U.S. geographic region where this trial was conducted, the study cohort was predominantly Hispanic and non-Hispanic white. This is an acknowledged limitation although unlikely alters mechanistic proof-of-concept. The randomized, placebo-controlled follow-up study will be conducted in multiple centers that capture the racial and ethnic diversity of North America thereby yielding more representative and generalizable results.

## 5. Conclusions

The present study adds to the growing body of literature suggesting that the consumption of isothiocyanates, concentrated in the *Brassica* family of cruciferous vegetables, enhances detoxification of environmental carcinogens, including benzene and acrolein, common to air pollution and tobacco smoke. The low toxicity and high compliance observed here and in other studies raise the promise of a tolerable, long term chemoprevention strategy against environmental carcinogenesis. Future studies will be facilitated by the availability of commercial dietary supplements, including the BSSE Avmacol^®^ studied here, with predictable concentrations of bioactive constituents, acceptable bioavailability, manufacture in accordance with GMP standards, and long shelf life at ambient temperature; this contrasts with the impractical cultivation, harvest, and manufacture of extracts from 3-day old broccoli sprouts, which are no longer available for clinical trials. Stepwise development of BSSE against tobacco-related carcinogenesis is warranted in high-risk populations. A multicenter, randomized, placebo-controlled trial evaluating the sustainability of benzene and acrolein detoxification by higher-dose BSSE over 12 weeks is now planned in otherwise healthy, heavy tobacco smokers (NCT05121051).

## Figures and Tables

**Figure 1 cancers-14-02129-f001:**
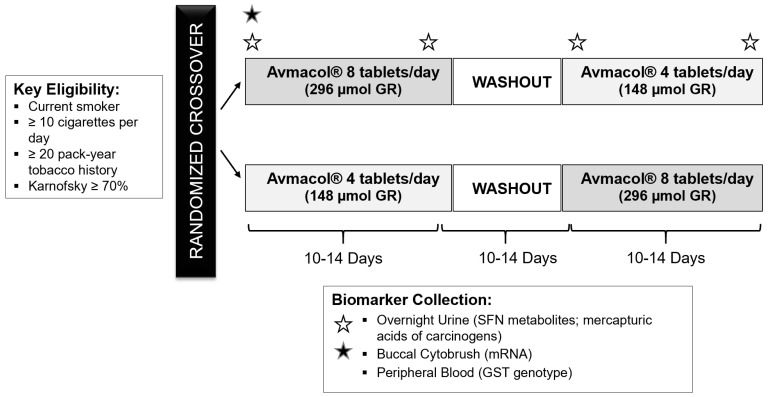
Study schema.

**Figure 2 cancers-14-02129-f002:**
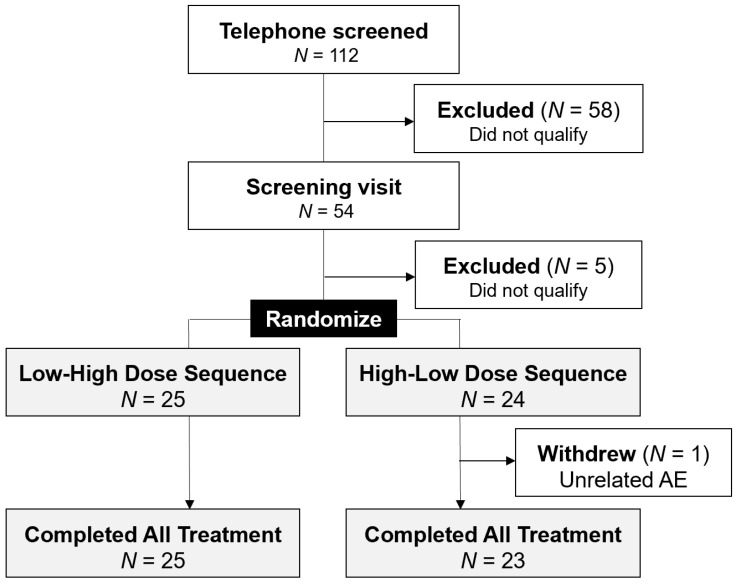
CONSORT diagram.

**Table 1 cancers-14-02129-t001:** Baseline characteristics.

Characteristic	*N* = 49 ^a^
Age, years: Median (range)	57 (37, 73)
Sex: *N* (%)	
Male	23 (47)
Female	26 (53)
Race: *N* (%)	
Black or African American	1 (2)
White	44 (90)
More than one	3 (6)
Unknown or not reported	1 (2)
Ethnicity: *N* (%)	
Hispanic or Latino	4 (8)
Not Hispanic or Latino	44 (90)
Unknown or not reported	1 (2)
Karnofsky performance status: *N* (%)	
90%	6 (12)
100%	43 (88)
Tobacco use: Median (range)	
Pack-years	36 (24, 60)
Cigarettes per day	20 (10, 30)
*GST* genotype: *N* (%)	
*GSTM1* null	23 (47)
*GSTT1* null	8 (16)

^a^ The denominator for *N* (%) calculations is 49.

**Table 2 cancers-14-02129-t002:** Urinary excretion of carcinogen metabolites.

Carcinogen (Metabolite) ^a^	BSSE Dose Level	Pre (95% CI) pmol/mgCr	Post (95% CI) pmol/mgCr	Post/Pre (95% CI)	*p*-Value ^d^
Benzene (SPMA)	Low	7.8 (6.3, 9.6) ^b^	9.0 (7.2, 11.3)	1.2 (1.0, 1.3) ^c^	0.05
	High	7.6 (6.2, 9.4)	9.1 (7.3, 11.3)	1.2 (1.0, 1.4)	0.04
Acrolein (3-HPMA)	Low	9934.5 (8008.0, 12,324.4)	11,032.1 (9161.1, 13,285.2)	1.1 (1.0, 1.3)	0.11
	High	9750.0 (7903.7, 12,027.6)	12,450.1 (10,658.8, 14,542.5)	1.3 (1.1, 1.5)	<0.01
Crotonaldehyde (3-HMPMA)	Low	10,962.6 (9026.8, 13,313.6)	11,482.0 (9677.5, 13,622.9)	1.1 (0.9, 1.2)	0.56
	High	10,808.6 (8947.3, 13,057.0)	12,795.1 (10,922.9, 14,988.1)	1.2 (1.02, 1.37)	0.02

^a^ SPMA, 3-HPMA, and 3-HMPMA were normalized to urine creatinine; ^b^ Geometric mean and the 95% CI derived from log-normal regression; ^c^ Ratio of the geometric mean between post and pre and the 95% CI derived from log-normal regression; ^d^ Derived from log-normal regression.

**Table 3 cancers-14-02129-t003:** Treatment-emergent adverse events.

Toxicity ^a^	Low Dose (*N* = 49)	High Dose (*N* = 49)	Both Arms (*N* = 49) ^b^	*p* Value ^c^
Abdominal Pain	1 (2%)	10 (20%)	10 (20%)	<0.01
Bloating	0 (0%)	1 (2%)	1 (2%)	NA ^d^
Diarrhea	3 (6%)	10 (20%)	11 (22%)	0.02
Loose Stool	7 (14%)	9 (18%)	12 (24%)	0.48
Flatulence	3 (6%)	7 (14%)	10 (20%)	0.21
Nausea	1 (2%)	0 (0%)	1 (2%)	NA
Vomiting	0 (0%)	1 (2%)	1 (2%)	NA
Weight Loss	0(0%)	1 (2%)	1 (2%)	NA

^a^ All reported AEs were Grade 1–2 according to NCI CTCAE v.4; ^b^ Participants who had AE in either or both arms; ^c^
*p*-value was derived using McNemar’s test; ^d^ NA: Not Applicable.

## Data Availability

Data supporting the reported results can be found at ClinicalTrials.gov, NCT03402230.

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
