# Peer review of "Randomized Crossover Trial Evaluating Detoxification of Tobacco Carcinogens by Broccoli Seed and Sprout Extract in Current Smokers"

_cancers, 2022, doi:10.3390/cancers14092129_

Round 1

Reviewer 1 Report

This study is an important trial that can contribute to the development of cancer research, and the experimental method is also appropriate. However, it requires some modifications.

  1. Please specify the software used for statistical analysis.

  1. Is there anything that should be noted from the perspective of cancer that the subject's race is mostly Caucasian?

  1. Viewing the characteristics of subjects (especially, race and ethnicity), are there any noted points to be aware of demonstrating for cancer?

  1. Please add a detailed description of the CONSORT Diagram (e.g. excluded criteria for screening visit).

Author Response

Reviewer 1

  1. Please specify the software used for statistical analysis.

Thank you, we have added this key detail to the manuscript. The software package used for statistical analysis was SAS 9.4 and is now specified in Section 2.4.

  1. Is there anything that should be noted from the perspective of cancer that the subject's race is mostly Caucasian?

This is an important question. The following has now been added to the discussion of the results.

“While representative of the tobacco-smoking population of Arizona, the U.S. geographic region where this trial was conducted, the study cohort was predominantly Hispanic and non-Hispanic white. This is an acknowledged limitation although unlikely alters mechanistic proof-of-concept. The randomized, placebo-controlled follow-up study will be conducted in multiple centers that capture the racial and ethnic diversity of North America thereby yielding more representative and generalizable results.”

  1. Viewing the characteristics of subjects (especially, race and ethnicity), are there any noted points to be aware of demonstrating for cancer?

 Please see response to #2 above.

  1. Please add a detailed description of the CONSORT Diagram (e.g. excluded criteria for screening visit).

We appreciate this recommendation to enhance clarity regarding the flow of patients through the trial, including exclusion. A legend has been added to the CONSORT diagram detailing reasons for exclusion at both screening steps.

Reviewer 2 Report

In this manuscript, Bauman et al., et al., conducted a Randomized Crossover Trial to evaluate the detoxification of tobacco Carcinogens by a commercially available extract prepared from Broccoli Seed and Sprout (BSSE) in smokers. The work is interesting.

Comments:

  1. Under section 2.2. “Avmacol… was used throughout the study; each tablet contained 16 mg (37 µmol) GR with SFN conversion of 6 mg (14 µmol)”. SFN has a molecular weight 177.29; 6mg SFN is : 6000/177.29 = 33.8 µmol. Not sure how did you get the conversion of “6mg SFN to 14 µmol”. Please explain.
  2. SFN Metabolites. It seems you did not detect SFN metabolites in plasma. Can you explain this?
  3. GSTM1 and GSTT1 Genotypes. Explain why GSTP1 not included in your genotyping.
  4. Explain if the doses (296, or 148 umol GR) can be achieved through normal diet.

Author Response

Reviewer 2

  1. Under section 2.2. “Avmacol… was used throughout the study; each tablet contained 16 mg (37 µmol) GR with SFN conversion of 6 mg (14 µmol)”. SFN has a molecular weight 177.29; 6mg SFN is : 6000/177.29 = 33.8 µmol. Not sure how did you get the conversion of “6mg SFN to 14 µmol”. Please explain.

Thank you very much for catching this typographical error! The manuscript has been corrected to “with SFN conversion of 6 mg (34 µmol).”

  1. SFN Metabolites. It seems you did not detect SFN metabolites in plasma. Can you explain this?

SFN metabolites were not measured in plasma, per study protocol, but rather in overnight urine in accordance with our prior studies as well as the standard approach utilized in the Qidong air pollution studies. This provides an estimate of bioavailability of BSSE as sulforaphane.

  1. GSTM1 and GSTT1 Genotypes. Explain why GSTP1 not included in your genotyping.

We performed GSTM1 and GSTT1 genotyping per protocol plan, which was modeled after prior studies including the Qidong air pollution study (Egner 2014) and the 2-phenethyl isothiocyanate trial in tobacco smokers (Yuan JM 2016). This is an acknowledged limitation, and GSTP1 polymorphisms will be considered in subsequent studies.

  1. Explain if the doses (296, or 148 umol GR) can be achieved through normal diet.

This important question has now been explored, with corresponding references, in the discussion section as follows:

“A frequent and important question in green chemoprevention, the use of plants or their simple extracts as cancer prevention agents, is whether a bioactive dose of protective phytochemicals could be achieved by normal diet. The GR content of market stage broccoli is highly variable, and is largely dependent upon genotype although environmental conditions such as field location, year, drought conditions, and disease pressure also play a role (51, 52). In a study of 31 fresh, uncooked (Brassica oleracea var. italica), market stage broccoli heads purchased at supermarkets in the Baltimore region, a mean GR content of 0.4 µmol (range 0.005 to 1.1 µmol) per gram was observed (53). On average, roughly 28 ounces of market stage broccoli would contain a similar GR content as higher-dose Avmacol® used in the present study. As broccoli sprouts contain 10-100 times the level of GR as mature plants, and are a rich source of myrosinase activity, their consumption may be more practical (32).”

Reviewer 3 Report

the study aims to explore the effect of broccoli seed and sprout extract on the detoxification of tobacco carcinogens in current smokers. The study is well executed. As a clinical study, it has been registered. No conflict of interest has been detected. Three points:

1) please provide graphical summary of the observed effects

2) please provide the exact list of ingredients of the supplement tested

3) maintake mushroom extractc and wasabi - please discuss their role in the observed effect

Author Response

Reviewer 3

Three points:

  1. please provide graphical summary of the observed effects

A graphical abstract has been developed and submitted with the revised manuscript.

  1. please provide the exact list of ingredients of the supplement tested

Thank you for this request. We have now included the ingredient list in the manuscript as follows: “The original Avmacol® formulation is comprised of GR-rich broccoli seed extract, freeze-dried broccoli sprouts for the myrosinase source, ascorbic acid, and inert excipients for tablet formation (carboxymethylcellulose, silicon dioxide, and microcrystalline cellulose).”

  1. maitake mushroom extract and wasabi - please discuss their role in the observed effect.

Thank you for the opportunity to clarify. The product used was the original Avmacol®, which does not contain maitake mushroom or wasabi. (The currently commercially available product Avmacol® ES contains maitake mushroom and wasabi.) As above, the precise list of ingredients is now provided in the manuscript.

Round 2

Reviewer 3 Report

The Authors clarified all the points